# Multi-Omics Techniques for Analysis Antifungal Mechanisms of Lipopeptides Produced by *Bacillus velezensis* GS-1 against *Magnaporthe oryzae* In Vitro

**DOI:** 10.3390/ijms23073762

**Published:** 2022-03-29

**Authors:** Yanhua Zhang, Meixi Zhao, Wei Chen, Huilin Yu, Wantong Jia, Hongyu Pan, Xianghui Zhang

**Affiliations:** College of Plant Science, Jilin University, Changchun 130012, China; yh_zhang@jlu.edu.cn (Y.Z.); zhaomq20@mails.jlu.edu.cn (M.Z.); chenw20@mails.jlu.edu.cn (W.C.); yuhl21@mails.jlu.edu.cn (H.Y.); jiawt21@mails.jlu.edu.cn (W.J.); panhongyu@jlu.edu.cn (H.P.)

**Keywords:** *Bacillus velezensis*, biocontrol, *Magnaporthe oryzae*, lipopeptide, genome sequencing, transcriptomics, metabolomics

## Abstract

*Magnaporthe oryzae* is a fungal pathogen that causes rice blast, a highly destructive disease. In the present study, the bacteria strain GS-1 was isolated from the rhizosphere soil of ginseng and identified as *Bacillus velezensis* through 16S rRNA gene sequencing, whole genome assembly, and average nucleotide identity analysis. *B. velezensis* strain GS-1 exhibited significant antagonistic activity to several plant fungal pathogens. Through whole genome sequencing, 92 Carbohydrate-Active Enzymes and 13 gene clusters that encoded for secondary metabolites were identified. In addition, strain GS-1 was able to produce the lipopeptide compounds, surfactin, fengycin, and plantazolicin. The inhibitory effects of lipopeptide compounds on *M. oryzae* were confirmed, and the antagonistic mechanism was explored using transcriptomics and metabolomics analysis. Differential expressed genes (DEGs) and differential accumulated metabolites (DAMs) revealed that the inhibition of *M. oryzae* by lipopeptide produced by GS-1 downregulated the expression of genes involved in amino acid metabolism, sugar metabolism, oxidative phosphorylation, and autophagy. These results may explain why GS-1 has antagonistic activity to fungal pathogens and revealed the mechanisms underlying the inhibitory effects of lipopeptides produced by GS-1 on fungal growth, which may provide a theoretical basis for the potential application of *B. velezensis* GS-1 in future plant protection.

## 1. Introduction

Rice is the most staple food crop and feeds three billion people in the world, especially in Asia. Wherever it is grown, rice is susceptible to a variety of pathogens. Rice blast, which is caused by *Magnaporthe oryzae*, is one of the most serious rice diseases worldwide, has occurred in >80 countries, and caused yield losses of nearly 30% every year [1]. Currently, the most common approaches to control rice blast are development of resistant cultivars and application of chemical fungicides. Generally, it takes a long time to develop disease-resistant varieties, and the physiological races of *M. oryzae* vary frequently. In addition, the overuse of chemical fungicides to control rice blast has caused fungicide resistance, environmental pollution, and threatens human health. Application of bioactive agents to control plant diseases is more attractive and has significant potential in the future.

It is well known that microorganisms with antifungal activity have been applied in many crop systems. For example, *Pseudomonas* spp., *Bacillus* spp., *Streptomyces* spp., *Trichoderma* spp., and *Chaetomium* spp. have been used to control rice blast in the field [2,3,4,5]. Among microbial candidates used for biocontrol, *Bacillus* species are most attractive because they can produce spores, and a series of bioactive compounds allow them to survive in adverse environmental conditions [6,7,8]. Moreover, *Bacillus* species have many functions, which include promotion of plant growth and stress resistance, improvement in soil fertility, and remediation of contaminated soil [9,10,11,12,13].

During the past several years, there has been an increasing amount of research on screening and applying *Bacillus* strains with antimicrobial activity. The endophytic *Bacillus subtilis* and *Bacillus megaterium* isolated from wheat inhibited vegetative growth and conidia germination of *Fusarium graminearum* significantly [14]. *B. subtilis* BS06 could effectively protect soybean root caused by *Fusarium oxysporum* [8]. The antagonistic strain *B. velezensis* SDTB038 achieved good control of potato late blight in greenhouses and fields and promoted potato plant growth [12]. *B. velezensis* CE 100 could simultaneously control *Phytophthora* root rot diseases and enhance growth of *Chamaecyparis obtusa* seedlings [13]. *Bacillus amyloliquefaciens* S170 and *B. pumilus* S9 isolated from nonrhizospheric rice soil showed significant antagonistic activity against *M. oryzae* in vitro and in leaf disk assays [15]. Zhou found *Bacillus cereus* YN917, obtained from a rice leaf with remarkable antifungal activity against *M. oryzae*, and exhibited multiple plant growth-promoting and disease prevention traits [16]. The cyclic lipopeptide fengycin BS155 from *B. subtilis* BS155 suppressed the growth of *M. oryzae* [17]. Bacillomycin D isolated from *B. velezensis* HN-2 injured the cell wall and cell membrane of the hyphae and spores of *Colletotrichum gloeosporioides* [18]. The alginate-coated, monolayer microcapsules of *B. velezensis* strain NH-1 has controlled *Fusarium* wilt in field experiments with 100% efficiency [19]. The *Bacillus safensis* JLS5 and *Bacillus tequilensis* JLS12 isolated from frozen soils of the soda saline-sodic land could effectively inhibit conidial germination and pathogenicity of the rice blast fungus [20]. The crude extracts of *Bacillus velezensis* of A6 and P42 significantly inhibited the rice pathogen *M. oryzae* and bacterial blight of pomegranate pathogen, *Xanthomonas axonopodis* pv. *Punicae* [21]. In conclusion, *Bacillus* strains have great potential and prospects for their application in the biological control of plant diseases.

In *Bacillus* spp., *B. velezensis* species included effective strains that had an extensive antimicrobial spectrum because they produced many antimicrobial agents, which included nonribosomal synthesis of lipopeptides (surfactin, bacillibactin, and fengycin), polyketides (macrolactin, bacillaene, and difficidin), thiazole/oxazole-modified microcin (plantazolicin), and carbohydrate metabolism [22,23,24,25,26,27]. At present, the main research focus is on the quick identification of antimicrobial agents in *Bacillus*. However, the underlying antagonistic mechanism of lipopeptide compounds was seldom explored.

Genome sequencing is a valuable strategy to identify secondary metabolite genes and potential biological control agents [28,29]. In addition, whole genome sequencing allows us to identify the taxonomic relationship among closely related *Bacillus* species more efficiently based on the 16S rRNA gene or average nucleotide identities (ANI) [30,31]. Moreover, through whole genome annotation, we can discover genes that are involved in secondary metabolite production, carbohydrate metabolism, and cell wall degradation [27,29]. High-throughput sequencing technologies have been used to explore complex biological and physiological processes of microbes. Transcriptomic analysis based on RNA sequencing is a reliable approach to profile genes that are expressed differently and for understanding the gene network [32]. Therefore, transcriptomics can provide a system-wide understanding of the complex biological processes of fungi.

In this study, *B. velezensis* GS-1 isolated from rhizosphere soil of ginseng showed significant antifungal activity against *M. oryzae* and seven other plant fungal pathogens. By using whole genome sequencing, the genes for secondary metabolites and CAZymes were identified. In addition, strain GS-1 was able to produce the lipopeptide compounds, surfactin, fengycin, and plantazolicin, which correlated with gene cluster analysis of secondary metabolites. Furthermore, the crude lipopeptide extracts from GS-1 exhibited strong antifungal activity against *M. oryzae* and *S. turcica*. In addition, the antifungal mechanism of lipopeptide compounds produced by *B. velezensis* GS-1 on *M. oryzae* was explored by transcriptomics and metabolomics analysis. All these results provide a theoretical basis for the development of biological control agents.

## 2. Results

### 2.1. Screening of Antagonistic Activity against Plant Pathogenic Fungi

More than 300 bacteria were isolated from the rhizosphere soil of ginseng, and the isolate GS-1 exhibited the highest degree of growth inhibition against *M. oryzae*. In addition, GS-1 showed significant antagonistic activity against seven other plant pathogens that included *Fusarium graminearum*, *Rhizoctonia solani*, *Botrytis cinerea*, *Cochliobolus heterostrophus*, *Cercospora zeae maydis*, *Sclerotinia sclerotiorum,* and *Setosphaeria turcica* (Figure 1). We concluded that GS-1 had a broad antagonistic spectrum, especially against ascomycetes.

### 2.2. Genomic Features of B. velezensis GS-1

The complete genome sequence of *B. velezensis* GS-1 was composed of a circular chromosome of 4,030,799 bp that included 4187 coding sequences (CDS) with an average length of 857.71 bp, and the average G + C content was 47.06%. In addition, the chromosome contained 27 rRNA, 87 tRNA, and four genomic islands, with an average length of 44,825.25 bp. Among the 4187 CDS, 4124 were annotated functionally, and the remaining 63 were hypothetical. From the predicted genes, 4124 (98.5%), 3571 (85.29%), 3057 (73.01%), 2181 (52.09%), and 2156 (51.49%) matched in the NR, SwissProt, COG, KEGG, and GO databases, respectively. Of these, 1465 genes were matched across all five databases (Figure 2).

### 2.3. Phylogeny of B. velezensis GS-1

The 16S rRNA gene sequences of *B. velezensis* GS-1 and 15 *Bacillus* species were used to construct a phylogenetic tree. From the phylogenetic tree, we found that *B. velezensis* Gs-1 was related most closely to the three *B. velezensis* strains, *B. velezensis* YB-130, *B. velezensis* FZB42, and *B. velezensis* CAU B946 (Figure 3). Furthermore, ANI analysis was conducted between GS-1 and the other 15 *Bacillus* species, and GS-1 showed highest values with *B. velezensis* CAU B946 (99.47%), followed by *B. velezensis* YB-130 (97.78%) and *B. velezensis* FZB42 (97.73%) (Figure 4). This was consistent with the result shown in the 16S rRNA phylogenetic tree. In addition, strain GS-1 also showed close relationships with *B. amyloliquefaciens* ALB 65 (97.71%) and *B. amyloliquefaciens* MT 45 (94.08%) in ANI analysis. The taxonomic identities of the GS-1 and the related strains were further confirmed by TYGS server. Based on genomic evidence, GS-1 was also confirmed as *B. velezensis*. But, the dDDH values from TYGS were a little lower than the values from ANI. For example, the ANI values of GS-1 compared with *B. velezensis* WRN014 and *B. velezensis* FZB42 were 99.24% and 97.73%, but only 94.5% and 80.3% in TYGS analysis. In addition, the percentage of G + C of GS-1 was 46.3%, the same as *B. velezensis* WRN014, and little lower than *B. velezensis* FZB42 (46.5%). These results indicated that GS-1 was closer to *B. velezensis* WRN014. Based on all the evidences mentioned above, strain GS-1 was classified as *B. velezensis.*

### 2.4. Analysis of CAZyme Genes in B. velezensis GS-1 Genome

*B. velezensis* GS-1 genome contained 92 CAZyme genes that included five auxiliary activity (AA) genes, six carbohydrate-binding module (CBM) genes, 20 carbohydrate esterase (CE) genes, 38 glycoside hydrolase (GH) genes, 20 glycosyl transferase (GT) genes, and three polysaccharide lyase (PL) genes (Figure 5A). Of the 92 CAZymes, glucosidase (GH4, GH13_29), chitinase (GH23, GH18), and glucanase (GH5_2, GH16) had potential antifungal activity (Figure 5B). In addition, 28 of 92 CAZyme genes had amino-terminal signal peptides, which indicated that they could be secreted out of the cytoplasmic membrane. The 28 CAZyme genes with signal peptides were grouped into five groups; the largest group was GH, which included 16 CAZymes with signal peptides, followed by CE, PL, CBM, and GT, which had six, three, two, and one CAZymes with signal peptides, respectively (Figure 5A). These results suggested that CAZymes in GS-1 played a significant role in antifungal activity.

### 2.5. Secondary Metabolic Related Genes of GS-1

The genome of *B. velezensis* GS-1 contained 13 gene clusters related to secondary metabolite biosynthesis, which covered 19.12% of the genome (Table 1). Among the 13 gene clusters, there were three NRPS (non-ribosomal peptide synthetase) gene clusters, including one gene cluster matched 82% similarity with surfactin encoding cluster, and the other two clusters showed 100% similarity with gene clusters for fengycin and bacillibactin synthesis; Three transAT-PKS (trans-acyl transferase polyketide synthetase) gene clusters, which consisted of one gene cluster had 100% similarity with gene cluster for bacillaene synthesis, one gene cluster had 100% similarity with gene cluster for difficidin synthesis and one gene cluster had 100% similarity with gene cluster for macrolactin H synthesis; Two terpene gene clusters, one PKS (polyketide synthetase) gene cluster showed 7% similarity with gene cluster for butirosin A/butirosin B synthesis; One thiopeptide gene cluster had 4% similarity with kijanimicin encoding cluster; One LAP (linear azoline-containing peptides) gene cluster had 91% similarity with gene cluster for plantazolicin synthesis; One T3PKS (Type III polyketide synthetase) gene cluster and one other gene cluster showed 100% similarity with bacilysin encoding cluster. Most of these genes were involved in antimicrobial production and were conserved in all *B. velezensis* strains. In NRPS gene clusters, the genes that encoded fengycin (*fen*F), surfactin (*srf*A), and bacillibactin (*dhb*F) were identified, and the predicted products of the PKS gene clusters were bacillaene (*bae*), difficidin (*dfn*), and macrolactin (*mln*). In addition, a gene cluster involved in terpene production and the biosynthetic genes yisP and sqhC were located. Moreover, the secondary metabolite plantazolicin, which is ribosomally synthesized, post-translationally modified peptides (RiPPs), and the core biosynthetic genes *ptn*B and *ptn*C were identified. The 13 gene clusters that encoded for secondary metabolites in the genome of *B. velezensis* GS-1 were all present in the genome of *B. velezensis* FJAT-450281, FZB42, and CAU B946.

### 2.6. Crude Lipopeptide Extracts from B. velezensis GS-1 Inhibited M. oryzae

Plate diffusion assays were used to test the inhibitory activity of crude lipopeptide extracts from *B. velezensis* GS-1 to plant pathogens; these extracts showed strong antagonistic ability to a variety of plant pathogens. The mycelium growth of *M. oryzae* and *S. turcica* were inhibited significantly, and the inhibition zone caused by CLE were 2.7 mm and 3 mm, respectively. *S. sclerotiorum* were inhibited slightly (Figure 6). MALDI-TOF MS was used to confirm the components in CLE. The first obvious peaks with *m/z* of 1030.7, 1044.7, 1072.8, and 1088.7 were predicted to belong to the surfactin family. The second peaks at a *m/z* of 1311.5, 1336.5, 1354.5, and 1376.5 were predicted to belong to the plantazolicin family. Lastly, peaks at a *m/z* of 1513.9, 1531.9, and 1560.8 were predicted to belong to the fengycin family (Figure 7).

### 2.7. Transcriptomics Analysis of the Effect of Lipopeptide Extracts on M. oryzae

To determine the global changes in gene expression, RNA-seq analysis was conducted to investigate the transcriptome of the effect of lipopeptide extracts on *M. oryzae*. DEGs were identified by comparing RNA sequence data from the lipopeptide extracts treatment with the control group. Genes with transcript levels log2FoldChange >1 and statistical significance (*p* < 0.05) were selected as DEGs. In total, 1689 DEGs were identified, and 657 DEGs were upregulated and 1032 DEGs were downregulated. To gain better insight into which biological processes were downregulated or upregulated by lipopeptide extracts treatment, Gene Ontology analysis was performed. Based on GO functional annotation, DEGs were categorized into three classes: cellular components, molecular functions, and biological processes. In category of biological processes, the most enriched were the oxidation-reduction process, transmembrane transporter, regulation of transcription, and pathogenesis. The most enriched biological processes in category of cellular component were nucleus, cytosol, cytoplasm, and integral component of membrane. However, the zinc ion binding, DNA binding, ATP binding, and RNA polymerase II transcription factor activity were mostly enriched in category of molecular function. The top 20 KEGG pathway enrichment analyses revealed that DEGs were involved mainly in the spliceosome, amino sugar and nucleotide sugar metabolism, valine, leucine and isoleucine degradation, fructose and mannose metabolism, purine metabolism, oxidative phosphorylation, tyrosine metabolism, starch and sucrose metabolism, and autophagy (Figure 8). Among the DEGs, some genes related to tyrosine metabolism (copper amine oxidase, MGG_13291), peroxisome (superoxide dismutase, MGG_07697), riboflavin metabolism (MFS transporter, MGG_00275), and ABC transporter (multidrug resistance protein 1, MGG_00239; cytochrome P450 monooxygenase, MGG_07593) were upregulated significantly. While, some genes involved in autophagy (sec17), DNA repair protein (RAD14, MGG_06208; RAD5, MGG_05032), and histone deacetylase (HDA1, MGG_01076; RPD3, MGG_05857) were downregulated obviously.

### 2.8. Metabolomics Analysis of the Effect of Lipopeptide Extracts on M. oryzae

A total of 980 metabolites were identified in *M. oryzae*. OPLS-DA was used to identify DAMs, and totally 158 DAMs were identified, among them, 43 metabolites were downregulated and 115 metabolites were upregulated. All DAMs (fold change ≥ 2 or ≤ 0.5), with the variable importance in the projection (VIP ≥ 1.0) between control and treatment are listed in Appendix A and Figure 9A. All 158 DAMs could be roughly grouped into 12 major classes, predominantly amino acid and its metabolites, organic acid and its derivatives, nucleotide and its metabolites, carbohydrate and its metabolites, glyceryl phosphatide, and fatty acid. The top 10 most differentially expressed metabolites are listed in Figure 9B. Among them, the upregulated metabolites include L-Histidine, 4-Methylhexanoic acid, 2,2-Dimethylpentanoic acid, *N*-methyl-4-aminobutyric acid, L-Histidinol, LPE (18:0/0:0), LPE (0:0/18:0), FFA (16:2) and Cytidine 5′-Dphosphocholine. The top 10 most downregulated metabolites are Protocatechuic Aldehyde, (R)-(-)-Mandelic acid, LTD4, α-Cyclohexylmandelic acid, PGD1, PGE1, Cinnamic acid, 3-(4-Hydroxyphenyl)-1-propanol, Ethyl *N*-acetyl-L-tyrosinate, and 3,4-Dimethylbenzoic acid.

One hundred and fifty-eight DAMs were enriched in 135 KEGG pathways, the most enriched pathways are Biosynthesis of cofactors, Biosynthesis of plant hormones, Biosynthesis of alkaloids derived from histidine and purine, and Galactose metabolism (Figure 10).

### 2.9. Integrated Analysis of Transcriptome and Metabolome in M. oryzae

To reveal the antimicrobial mechanism of lipopeptide, integrated transcriptome and metabolome analysis were conducted. After mapping molecular objects to KEGG pathway, 43 pathways were found to be regulated by both DEGs and DAMs (Figure 11). Among them, the most enriched pathways are Purine metabolism and Amino sugar and nucleotide sugar metabolism, followed by Pyrimidine metabolism, Tyrosine metabolism, Fructose and mannose metabolism, and so on. Among them, most DEGs and DAMs involved in pentose and glucuronate interconversions were upregulated. In addition, most genes and metabolites involved in glutathione and glycerol lipid metabolism were also upregulated. In contrast, most of genes and metabolites participated in arginine biosynthesis, alanine, aspartate and glutamate metabolism, and lysine degradation were downregulated.

## 3. Discussion

Biological control, which has become an effective measure in recent years, is one of the safest and environmentally friendly methods for plant disease control. Among biological control bacteria, *Bacillus* spp. were believed to be effective [33]. In recent years, *Bacillus* spp. have provided a useful strategy for management of *F. graminearum*, *F. oxysporum*, and *Xanthomonas axonopodis* [8,12,21,24,34]. In addition, *Bacillus* spp. produce many antifungal components, such as lipopeptides, β-1,4-glucanase, and chitinase [26,27,35,36,37,38]. These antifungal components had a strong antagonistic activity to a series of phytopathogens. In our study, one bacteria strain was isolated from the rhizosphere soil of ginseng that exhibited extreme antifungal activity to phytopathogens, especially to *M. oryzae*. Many *Bacillus* strains, which included *Bacillus amyloliquefaciens*, *Bacillus subtilis,* and *B. velezensis,* have inhibitory activities against mycelial growth or spore germination [24,34,38]. For example, endophytic *B. velezensis* strain B-36 inhibited *F. oxysporum* growth [34], *B. amyloliquefaciens* FZB42 inhibited *F. graminearum* growth and mycotoxin biosynthesis [38], and *B. subtilis* YB-05 showed significant inhibitory activity to *Gaeumannomyces graminis* [39]. However, seldom *Bacillus* isolates have antagonistic activity against *M. oryzae*, a hemibiotrophic fungus that causes rice blast, which is a serious threat to global rice production. Our research demonstrated that *B. velezensis* GS-1 inhibited mycelial growth of *M. oryzae* significantly. This presents a new solution and a solid foundation for the control of rice blast.

To further explore the antagonistic mechanism of GS-1, whole genome sequencing was conducted. Several gene clusters that encoded secondary metabolites were found in the genome sequencing of *B. velezensis* GS-1, which included surfactin, kijanimicin, plantazolicin, butirosin A/butirosin B, macrolactin H, bacillaene, fengycin, difficidin, bacillibactin, and bacilysin. Among these secondary metabolites, some exhibited antagonistic activity against filamentous fungi, such as fengycin [17,38,40], bacilysin, difficidin [41], bacillibactin [42], and macrolactin H [43]. In addition to these secondary metabolites, there were also other compounds with antifungal activity that were absent in *B. velezensis* GS-1. Lantibiotics encoded by the lanthipeptide gene cluster that was produced by *B. velezensis* RC 218 and YB-130 reduced *Fusarium* head blight and deoxynivalenol accumulation, but the lanthipeptide gene cluster was not found in our study [24,44]. In contrast, plantazolicin, which is a ribosomally synthesized peptide found in GS-1, was seldom found in other *B. velezensis* isolates. Plantazolicin was isolated originally from *B. velezensis* FZB42, and it displayed antibacterial activity toward closely related Gram-positive bacteria. We confirmed that the strong antifungal activity of GS-1 was related to the fact that it contained so many secondary metabolite gene clusters and antimicrobial secreted CAZymes.

In our study, the acid precipitation method was used to separate lipopeptides from cell-free fermentation samples. The MALDI-TOF result indicated that both surfactin and fengycin were present in GS-1, as was plantazolicin. This result complemented the analysis of genome sequencing and may explain why GS-1 had such a strong antifungal activity. To further explore the underlying antagonistic mechanism of lipopeptides produced by GS-1, transcriptomics was conducted. In this study, 1689 DEGs were identified in analyses of transcriptomes, and several genes and pathways that may be involved in lipopeptides detoxification in *M. oryzae*. In phytopathogenic fungi, endoglucanase, glycoside hydrolase, and xylanase always play an important role in development and virulence. For instance, glycoside hydrolase PXO_03177 27 in *Xanthomonas oryzae*, Pgchi3/4 and Pgchi 5/6 in *Pythium guiyangense* [45] are necessary for their pathogenicity on host plants. PsXEG1, an apoplastic endoglucanase secreted by *Phytophthora sojae* is essential for its full virulence on soybean [46]. The endo-β-1,4-xylanase BcXyn11A in *Botrytis cinerea* is responsible for plant cell-wall degrading during infection of hosts [47]. In transcriptomics analysis, several genes (i.e., MGG_07686, MGG_10972, MGG_14602, MGG_14903, MGG_07955, and MGG_10602) annotated as endoglucanase, xylanase, and glycoside hydrolase were downregulated in *M. oryzae* when treated with lipopeptides. All these results indicated that these kinds of enzymes were critical for development and virulence in pathogens. Therefore, we speculated that the reduced growth of *M. oryzae* when treated with lipopeptides was related to downregulation of these endoglucanase, xylanase and hydrolases.

The histone deacetylase and acetyltransferase of fungi is critical for growth, development and DNA damage repair. In this study, two histone deacetylase HDA1 (MGG_01076) and RPD3 (MGG_05857), and a histone acetyltransferase GCN5 (MGG_03677) were downregulated in transcriptomics analysis. Therefore, we speculated that the histone deacetylase and acetyltransferase were inhibited significantly in *M. oryzae* when exposure to lipopeptides and caused restrained growth. A histone deacetylase, *MoHDA1* regulates asexual development and virulence in the rice blast fungus *M. oryzae* [48]. In the phytopathogenic fungus *Alternaria alternate*, deletion of *HDA1*, *RPD3*, and *GCN5* resulted in reduced growth and conidiation [48]. Moreover, Δ*Gcn5* showed severe defects in resistance to DNA-damaging agents, indicating the critical role of HATs and HDACs in DNA damage repair [49]. In addition, several DNA repair family proteins were downregulated in our transcriptomics analysis, included MGG_04561, MGG_05032, and MGG_06208. These results implied that the lipopeptides may inhibit the growth and DNA damage repair of *M. oryzae* by downregulated the expression of genes related to histone deacetylase, histone acetyltransferase and DNA repair proteins.

On the contrary, several genes were upregulated in *M. oryzae* in response to lipopeptides. The first group is cytochrome P450, a battery of cytochrome P450 genes were upregulated, included MGG_04345, MGG_04911, MGG_12496, MGG_05215, MGG_08947, and MGG_07593. It has been reported that P450-mediated detoxification is one of the most critical mechanisms in fungi and insect, and many P450 genes were upregulated when treated with fungicide, insecticide, and other chemicals [50,51,52,53,54,55]. The intracellular redox balance of fungi is critical for signal transduction and development, especially, when fungi are faced with stress. In this study, one oxidoreductase (MGG_03812) was upregulated more than 1000 folds, and two other oxidoreductases (MGG_10710 and MGG_13440) were also upregulated. In addition, two NADPH oxidase genes (MGG_03823 and MGG_08297) showed the same regulation trend. These results indicated that the redox enzymes may played a vital role in *M. oryzae* in response to lipopeptide treatment. In several fungal species, ATP-binding cassette (ABC) or major facilitator superfamily (MFS) transporters played a vital role in resistance to various toxic compounds, such as secondary metabolites, antibiotics, and fungicides [56]. In our transcriptomics analysis, one MFS transporter gene (MGG_10131) in *M. oryzae* was upregulated when treated with lipopeptide. In our previous study, two MFS transporters were upregulated in *C. heterostrophus* under isothiocyanates treatment. MFS transporter also necessary for isothiocyanates degradation and virulence in *Botrytis cinerea*. These results suggested that MFS transporter provided tolerance to toxic compounds in fungi.

In addition, we also analyzed the inhibitory mechanism of lipopeptide from the perspective of metabolite changes using metabolome technique. Several metabolites involved in glutathione metabolism and ABC transporters pathways were upregulated which known as key factors in detoxify. Two pathways of *M. oryzae* related to carbohydrate metabolic (citrate cycle and galactose metabolism) were affected by lipopeptide treatment. Energy and carbon metabolism always considered to be the target of a series of fungicides. Once energy production was blocked, the mycelium growth of *M. oryzae* will be inhibited, and this would be explaining why the mycelium growth of *M. oryzae* was inhibited when exposure to lipopeptide extract from *B. velezensis* GS-1. In the future, we will focus on the separation of surfactin, fengycin, and plantazolicin produced by GS-1, and we will conduct antifungal tests for each compound.

## 4. Materials and Methods

### 4.1. Strains

*B. velezensis* GS-1 was isolated from the rhizosphere soil of ginseng in Jingyu, China. Generally, GS-1 was cultivated on Luria-Bertani (LB) solid/liquid medium at 37 °C. The fungal pathogens included *Magnaporthe oryzae*, *Fusarium graminearum*, *Rhizoctonia solani*, *Botrytis cinerea*, *Cochliobolus heterostrophus*, *Cercospora zeae-maydis*, *Sclerotinia sclerotiorum,* and *Setosphaeria turcica* used in this study were cultivated on PDA at 25 °C.

### 4.2. In Vitro Antagonistic Activity of B. velezensis GS-1

More than 300 bacterial strains were screened for their antagonistic activity against fungal pathogens by the dual culture method. A 5 mm diameter fresh fungal disc was placed in the center of a potato dextrose agar (PDA) plate. Then, 20 μL of 10^8^ CFU/mL *B. velezensis* GS-1 broth was inoculated on the left and right sides of the fungal disc at 3 cm distance. PDA plates that were inoculated only with fungal pathogens were used as negative control. After 5 d of incubation at 25 °C, inhibition zones and colony diameters were measured. All experiments included three replications and were repeated three times.

### 4.3. DNA Extraction, Genome Sequencing and Assembly

High-quality genomic DNA was extracted from GS-1 using a modified CTAB method. The quality and quantity of the extracted DNA were examined using a NanoDrop 2000 spectrophotometer (Thermo Fisher Scientific, Waltham, MA, USA), Qubit dsDNA HS Assay Kit on a Qubit 3.0 Fluorometer (Life Technologies, Carlsbad, CA, USA) and electrophoresis on a 0.8% agarose gel. Qualified genomic DNA was fragmented using G-tubes (Covaris, Woburn, MA, USA) and then end-repaired to prepare SMRTbell DNA template libraries (with a fragment size > 10 kb using the Blue Pippin system) according to the manufacturer’s instructions (Pacific Biosciences, Menlo Park, CA, USA). Library quality was analyzed by Qubit, and average fragment size was estimated using an Agilent 2100 Bioanalyzer (Agilent, Santa Clara, CA, USA). SMRT sequencing was performed using a Pacific Biosciences Sequel II sequencer (Frasergen Bioinformatics Co., Ltd., Wuhan, China). The PacBio reads were de novo assembled using Microbial Assembly (smrtlink8), HGAP4, and Canu (v.1.6) software. The depth of genome coverage was analyzed using the pbalign (BLASR, v.0.4.1) tool. The assembled genome sequence of strain GS-1 was deposited in the NCBI GenBank with accession number CP072791. A circular map of the genome of strain GS-1 was obtained using Circos (v.0.64).

### 4.4. Genome Annotation of GS-1

The complete genome of GS-1 was annotated using Glimmer (v.3.02). Ribosomal RNA genes were identified with RNAmmer (v.1.2), and tRNA genes were verified by tRNAscan-SE (v.2.0). A whole genome BLAST search (*E*-value threshold of 1 × 10^-5^) was performed against the Non-Redundant protein database (NR), SwissProt, COG (Cluster of Orthologous Groups of proteins), KEGG (Kyoto Encyclopedia of Genes and Genomes), and GO (Gene Ontology). We picked the hit with the highest score as the final annotation.

### 4.5. Identification of GS-1

The 16S rRNA sequence of GS-1 and 15 related Bacillus species (i.e., *B. velezensis* FZB42, *B. velezensis* YB-130, *B. velezensis* CAU B946, *B. subtilis* 168, *B. subtilis* Ps832, *B. subtilis* GQJK2, *B. amyloliquefaciens* ALB65, *B. amyloliquefaciens* MT45, *B. pumilus* ZB201701, *B. pumilus* TUAT1, *B. licheniformis* ATCC 14580, *B. licheniformis* SRCM103583, *B. altitudinis* W3, *B. altitudinis* PAE4, and *B. altitudinis* Lc5) were used to construct the phylogenetic tree with MEGA 7.0 using the Neighbor Joining method. In addition, ANI (Average Nucleotide Identity) analysis between GS-1 and the other 15 Bacillus strains that used whole genome sequences was calculated using OAT (0.90). For a reliable identification of the strains, the Type (Strain) Genome Server (TYGS) (at https://tygs. dsmz. de 15 March 2022) was employed, and the genomic DNA G + C content and Digital DNA-DNA hybridization (dDDH) values were calculated.

### 4.6. Analysis of CAZymes and Secondary Metabolic Genes

Annotated gene sequences from the genome of GS-1 were blasted using the carbohydrate active enzyme (CAZy) database (dbCAN-HMMdb-V7) by HMMER (v.3.1b2), and the *E*-value threshold was 1 × 10^-5^. Secondary metabolic gene clusters of *B. velezensis* GS-1 were predicted using anti-SMASH (v.5.1.1).

### 4.7. Preparation of Crude Lipopeptide Extracts from B. velezensis GS-1

Inoculum of *B. velezensis* GS-1 in LB medium was inoculated into 200 mL of Landy medium, followed by incubation at 30 °C for 2 d with shaking at 200 rpm. The fermentation sample was centrifuged for 20 min at 12,000 rpm, and the supernatant was collected. The pH of the supernatant was adjusted to 2 with HCl and then stored at 4 °C. After 12 h, the supernatant was centrifuged for 20 min at 12,000 rpm to collect the crude lipopeptide extracts. The extracts were dissolved with methanol three times, and the insoluble extracts were filtrated out. A rotary evaporator was used to evaporate the methanol to harvest the crude lipopeptides.

The antagonistic activity of crude lipopeptide extracts from *B. velezensis* GS-1 was determined by a plate diffusion method. A piece of mycelium plug was placed into the center of a 90-mm Petri plate with PDA, then two holes were punched with a sterile cork borer, and a volume of the crude lipopeptide was introduced into the well. The plates were incubated for 12 d at 25 °C, and antagonistic activity was tested.

### 4.8. RNA Extraction and Transcriptomics Analysis

For RNA preparation, total RNA of mycelia was extracted from the lipopeptide treatment group and the control group using Trizol reagent (Thermo fisher, 15596018) following the manufacturer’s procedure. The total RNA quantity and purity were analyzed with Bioanalyzer 2100 and RNA 6000 Nano LabChip Kit (Agilent, Palo Alto, CA, USA, 5067-1511), high-quality RNA samples with RIN number > 7.0 were used to construct sequencing library. After total RNA was extracted, mRNA was purified from total RNA (5 μg) using Dynabeads Oligo (dT) (Thermo Fisher, San Jose, CA, USA) with two rounds of purification. Following purification, the mRNA was fragmented into short fragments using divalent cations under elevated temperature (Magnesium RNA Fragmentation Module (NEB, cat. e6150, USA) under 94 °C 5–7 min). Then the cleaved RNA fragments were reverse-transcribed to create the cDNA by SuperScript™ II Reverse Transcriptase (Invitrogen, cat. 1896649, Carlsbad, CA, USA), which were next used to synthesis U-labeled second-stranded DNAs with *E. coli* DNA polymerase I (NEB, cat.m0209, Beverly, MA, USA), RNase H (NEB, cat.m0297, USA) and dUTP Solution (Thermo Fisher, cat. R0133, USA). An A-base was then added to the blunt ends of each strand, preparing them for ligation to the indexed adapters. Each adapter contained a T-base overhang for ligating the adapter to the A-tailed fragmented DNA. Dual-index adapters were ligated to the fragments, and size selection was performed with AMPureXP beads. After the heat-labile UDG enzyme (NEB, cat.m0280, USA) treatment of the U-labeled second-stranded DNAs, the ligated products were amplified with PCR. The average insert size for the final cDNA libraries were 300 ± 50 bp. At last, we performed the 2 × 150 bp paired-end sequencing (PE150) on an Illumina Novaseq™ 6000 following the vendor’s recommended protocol. Clean reads were filtered from raw data, and low-quality reads were removed. We aligned reads of all samples to the *M. oryzae* reference genome using HISAT2 (https://daehwankimlab.github.io/hisat2/, 15 March 2022, version: hisat2-2.0.4) package. Genes differential expression analysis was performed by DESeq2 software between two different groups. The genes with the parameter of false discovery rate (FDR) below 0.05 and absolute fold change ≥ 2 were considered differentially expressed genes. Differentially expressed genes were then subjected to enrichment analysis of GO functions and KEGG pathways.

### 4.9. Verification of RNA-Seq

RT-qPCR was used to verify the results of RNA-seq, and 10 DEGs (five upregulated and five downregulated) were selected. Total RNA was isolated with a TransZol Up Plus RNA kit (TransGen Biotech Co., Ltd., Beijing, China) and quantified using a NanoDrop2000 spectrophotometer. Real-Time PCR was performed with a PrimePro 48 real-time detection system (TECHNE, Abingdon, UK). Each experiment was repeated three times.

### 4.10. Metabolome Analysis

In order to determine the changes of metabolites in *M. oryzae* exposure to lipopeptide compounds, metabolome was performed. Approximately 20 mg of mycelium powder were ground in liquid nitrogen, and mixed with 400 μL solution (Methanol:Water = 7:3, *v*/*v*) containing internal standard, subsequently shake at 1500 rpm for 5 min. The sample was placed in liquid nitrogen for 5 min and on the dry ice for 5 min, and then thawed on ice and vortexed for 2 min. This freeze-thaw circle was repeated three times in total. The sample was centrifuged at 12,000 rpm for 10 min at 4 °C, 300 μL of supernatant was collected and placed in −20 °C for 30 min, followed by centrifugation at 12,000 rpm for 3 min, and a 200 μL aliquots of supernatant were transferred for LC-MS analysis. An LC-ESI-MS/MS system (UPLC, ExionLC AD, https://sciex.com.cn/; 15 March 2022 MS, QTRAP^®^ System, https://sciex.com/, 15 March 2022) was used for metabolite detection. The analytical conditions were as follows, UPLC: column, Waters ACQUITY UPLC HSS T3 C18 (1.8 μm, 2.1 mm × 100 mm); column temperature, 40 °C; flow rate, 0.4 mL/min; injection volume, 2μL; solvent system, water (0.1% formic acid): acetonitrile (0.1% formic acid); gradient program, 95:5 *v/v* at 0 min, 10:90 *v/v* at 11.0 min, 10:90 *v/v* at 12.0 min, 95:5 *v/v* at 12.1 min, 95:5 *v/v* at 14.0 min. The ESI source operation parameters were as follows: source temperature 500 °C; ion spray voltage (IS) 5500 V (positive), −4500 V (negative); ion source gas I (GSI), gas II (GSII), curtain gas (CUR) was set at 55, 60, and 25.0 psi, respectively; the collision gas (CAD) was high. Significantly regulated metabolites between groups were determined by VIP ≥ 1 and absolute Log2FC (fold change) ≥ 1. Identified metabolites were mapped to KEGG pathway database.

## 5. Conclusions

In current study, a bacteria strain GS-1 was isolated and identified as *Bacillus velezensis* through 16S rRNA gene sequencing and whole genome sequencing. *B. velezensis* GS-1 and the lipopeptide compounds extracted from GS-1 showed significant antagonistic activity to phytopathogens. In order to explore the underlying antagonistic mechanism of lipopeptide compounds, the transcriptomics and metabolomics analysis were conducted, and several pathways were found may play an important role in detoxify in *M. oryzae*. The results may provide a theoretical basis for the potential application of *B. velezensis* GS-1 in future plant protection.

## Figures and Tables

**Figure 1 ijms-23-03762-f001:**
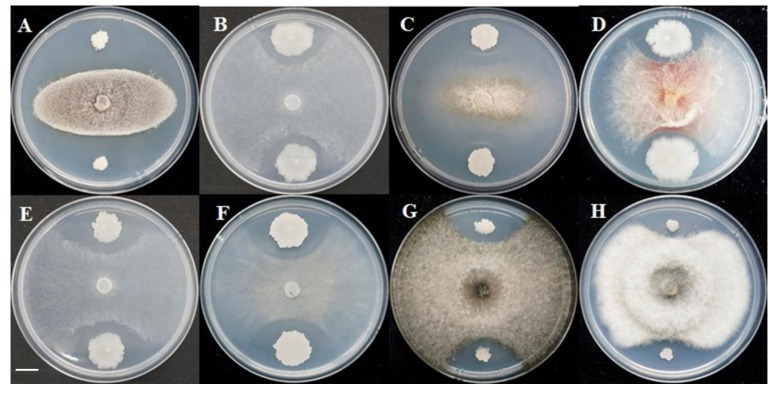
Antagonistic activity of B. velezensis GS-1 against plant pathogenic fungi. (**A**) Magnaporthe oryzae, (**B**) Rhizoctonia solani, (**C**) Botrytis cirerea, (**D**) Fusarium graminearum, (**E**) Sclerotinia sclerotiorum, (**F**) Cercospora zeae, (**G**) Setosphaeria turcica, (**H**) Cochiliobolus heterostrophus. Scale bar: 1 cm.

**Figure 2 ijms-23-03762-f002:**
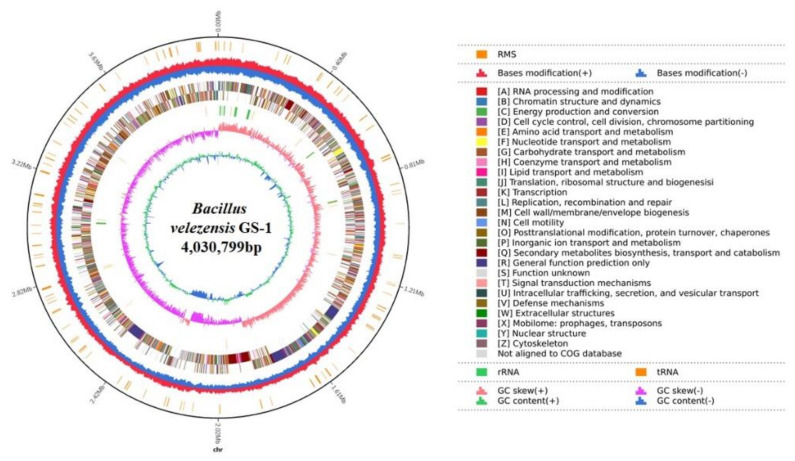
Circular genome map of *B. velezensis* GS-1. From the innermost to the outermost: ring 1 for GC content, ring 2 for GC skew, ring 3 for distribution of rRNAs (green) and tRNAs (brown), ring 4 for COG classifications of protein-coding genes on the forward strand and reverse strand, ring 5 for restriction modification system, and ring 6 for genome size (black line).

**Figure 3 ijms-23-03762-f003:**
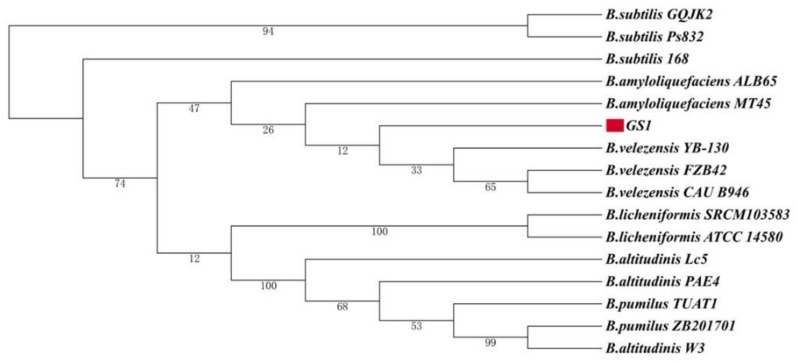
Phylogenetic tree of GS-1 and 15 other *Bacillus* species based on 16S rRNA sequence analysis. The red color indicated the strain GS-1 isolated in this study.

**Figure 4 ijms-23-03762-f004:**
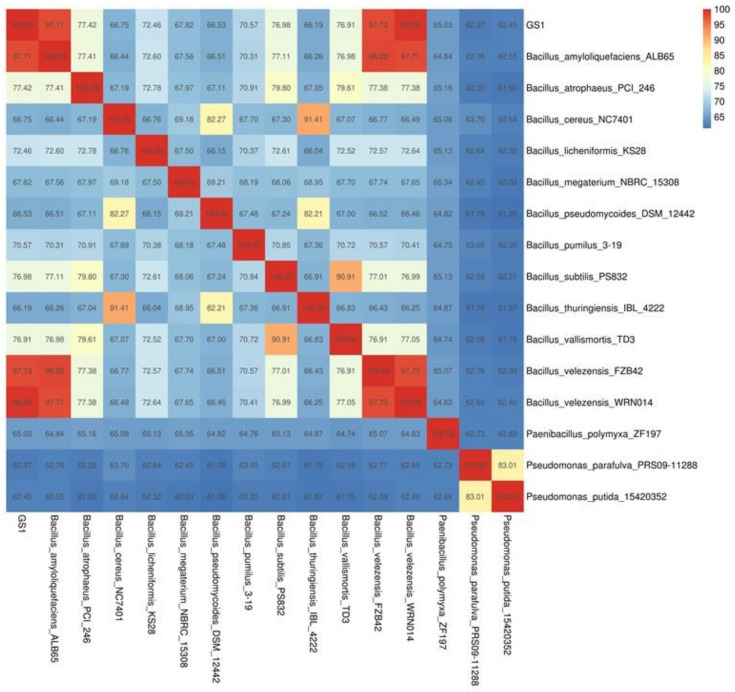
Heatmap of pairwise average nucleotide identity (ANI) values for whole genomes of GS-1 and 15 other *Bacillus* species.

**Figure 5 ijms-23-03762-f005:**
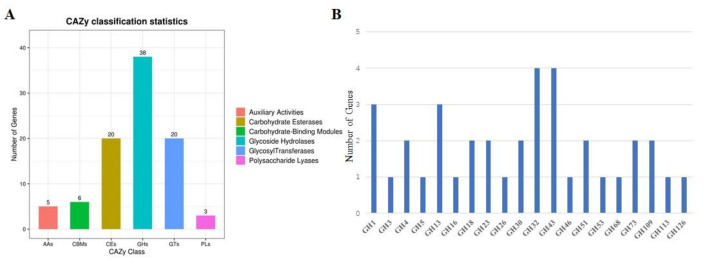
CAZy genes classification and glycoside hydrolase (GH) distribution in strain GS-1. (**A**) CAZy gene classification in GS-1 genome; (**B**) GH genes distribution in GS-1 genome.

**Figure 6 ijms-23-03762-f006:**
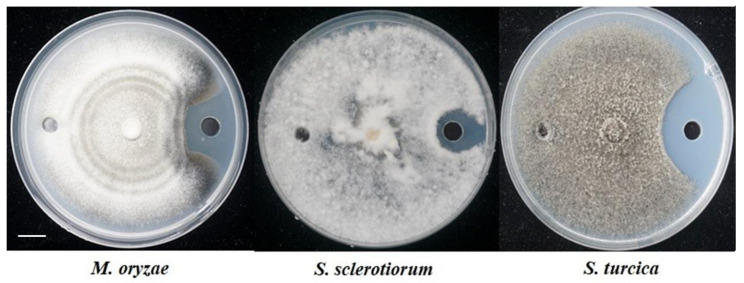
Antagonistic activity of crude lipopeptide extract from *B. velezensis* GS-1 against indicated plant pathogenic fungi. Scale bar: 1 cm.

**Figure 7 ijms-23-03762-f007:**
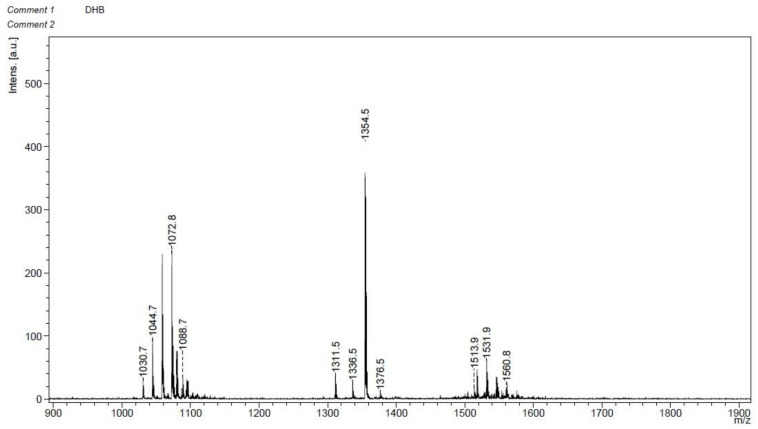
Matrix-assisted laser desorption ionization time-off light (MALDI-TOF) mass spectrometry (MS) spectra of the crude lipopeptide extracts.

**Figure 8 ijms-23-03762-f008:**
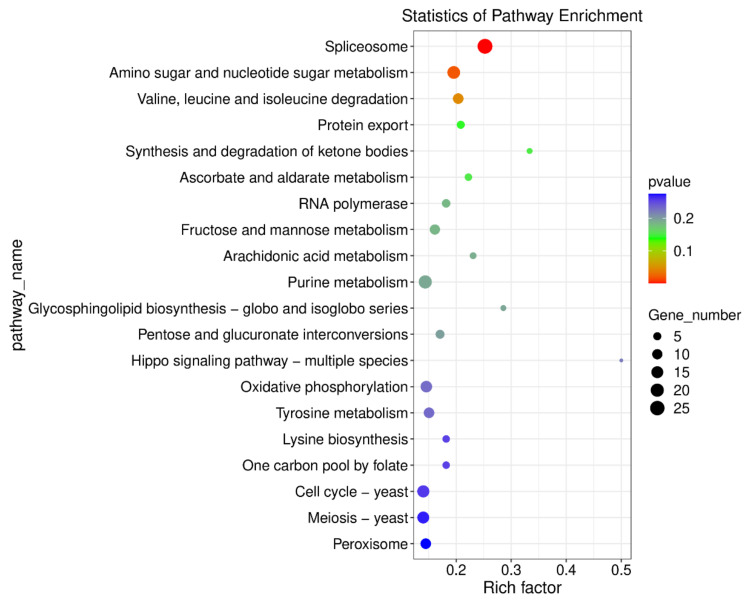
KEGG enrichment scatter plot. The ordinate represents the KEGG pathway, the abscissa represents the Rich factor. The larger the Rich factor, the greater the enrichment. The larger the point, the greater the number of differential genes enriched in the pathway. The redder the color of the dots, the more significant the enrichment.

**Figure 9 ijms-23-03762-f009:**
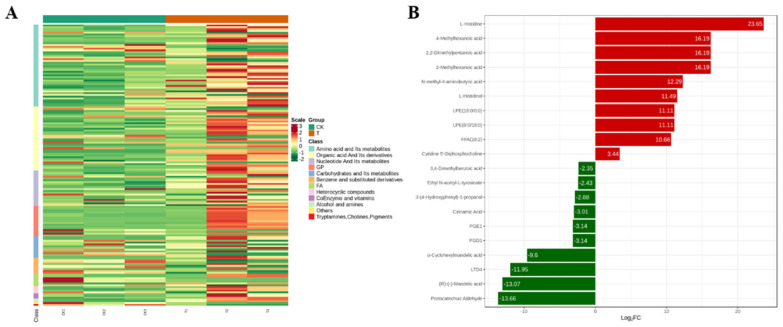
(**A**) Heatmap of differentially accumulated metabolites. (**B**) Top 10 DAMs in control and treatment.

**Figure 10 ijms-23-03762-f010:**
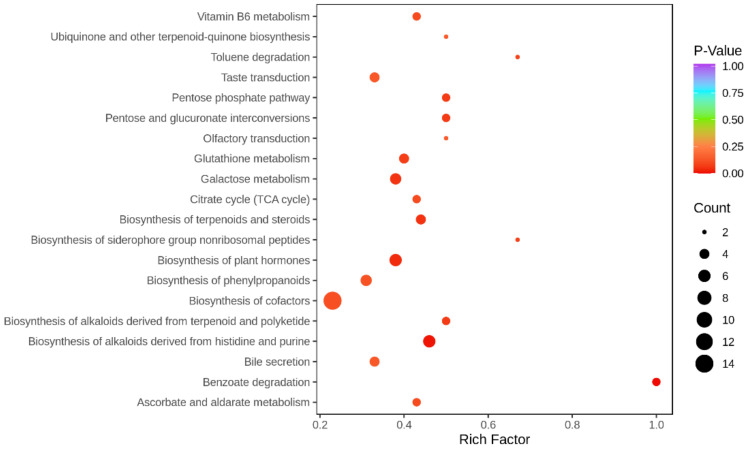
Differential accumulated metabolites KEGG enrichment. The size of the points in the graph represents the number of distinct significant metabolites enriched into the corresponding pathway. The more *p*-Value approaches to 0, the more significant the enrichment is.

**Figure 11 ijms-23-03762-f011:**
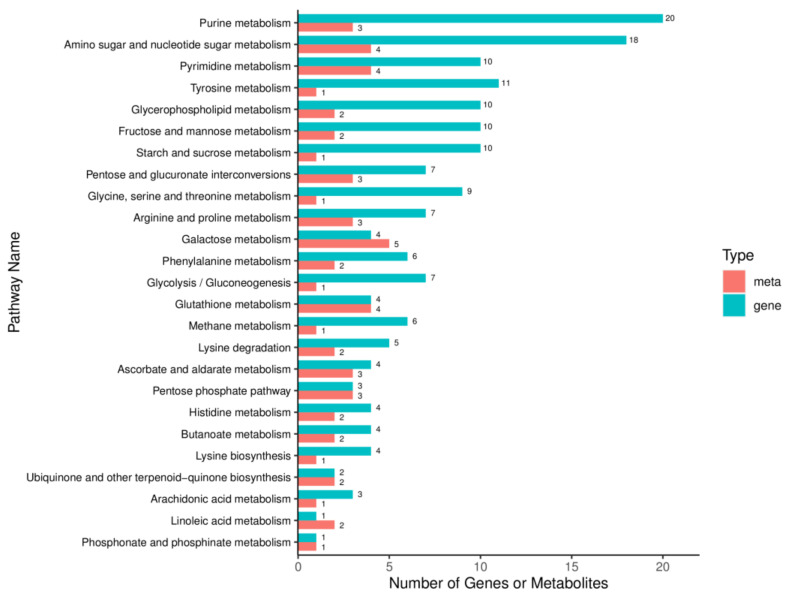
KEGG pathways significantly enriched in DEGs and DAMs.

**Table 1 ijms-23-03762-t001:** Putative gene clusters that encoded for secondary metabolites in GS-1.

Gene Clusters	Types	Genome Locations	Most Similar Known Clusters	Similarity
Cluster 1	NRPS	304,866–369,675	surfactin	82%
Cluster 2	thiopeptide, LAP	585,385–614,269	kijanimicin	4%
Cluster 3	LAP	698,756–720,938	plantazolicin	91%
Cluster 4	PKS-like	930,568–971,812	butirosin A/butirosin B	7%
Cluster 5	terpene	1,056,704–1,074,038		
Cluster 6	transAT-PKS	1,372,752–1,460,872	macrolactin H	100%
Cluster 7	transAT-PKS, T3PKS, transAT-PKS-like, NRPS	1,680,080–1,789,798	bacillaene	100%
Cluster 8	NRPS, transAT-PKS, betalactone	1,847,452–1,983,301	fengycin	100%
Cluster 9	terpene	2,006,887–2,028,770		
Cluster 10	T3PKS	2,097,450–2,138,550		
Cluster 11	transAT-PKS-like, transAT-PKS	2,391,006–2,497,179	difficidin	100%
Cluster 12	NRPS, bacteriocin	3,126,907–3,178,699	bacillibactin	100%
Cluster 13	other	3,691,456–3,732,874	bacilysin	100%

## Data Availability

Data available online: https://www.ncbi.nlm.nih.gov/nuccore/CP072791.1/.

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
