# Peer review of "Multi-Omics Techniques for Analysis Antifungal Mechanisms of Lipopeptides Produced by Bacillus velezensis GS-1 against Magnaporthe oryzae In Vitro"

_ijms, 2022, doi:10.3390/ijms23073762_

Round 1
Reviewer 1 Report
I suggest to change article title to
Multi-Omics techniques for analysis antifungal mechanisms of lipopeptides produced by Bacillus velezensis GS-1 against Magnaporthe oryzae 3 in vitro
Comments
I think the study could give important information to the scientific community.
We encourage you to prepare a high-quality picture as the graphic abstract which could be a single, concise, pictorial and visual summary of the main findings of the article.
Name of all microbes should be italicized. Your manuscript must re-format based on MDPI style.
Please include some of the latest research findings and updated reviews during 2021-2022 needed in the introduction and discussion parts. Also, in introduction, insert references for text 74-83.
Furthermore, all figures must be updated to be error-free and of sufficient quality for publishing in microorganisms. Text inside figures must be readable.
The conclusion you have provided is quite brief and you must provide sufficient feedback on the main objectives of your study, serrate text from 376-386 under subtitle Conclusion
The manuscript must be revised completely before it can be submitted and reviewed again.
An English language revision is recommended since frequently the use of the grammar and the syntax results are quite weird.
Regards,
Author Response
Dear reviewer,
So appreciated for your comments. It's really helpful for improving the quality of the manuscript. We have made a lot of revisions in the manuscript, please see the attachment.
Best.
Xianghui Zhang

Reviewer 2 Report
Dear authors,
In the present study “Analysis of antifungal mechanism of lipopeptide compounds produced by Bacillus velezensis GS-1 on Magnaporthe oryzae using Omics technology” by Zhang et al., the authors describe the isolation of bacterial strain GS-1 from the rhizosphere soil of ginseng. The bacterium was identified as Bacillus velezensis through 16S rRNA gene sequencing, whole genome assembly, and average nucleotide identity analysis. GS-1 exhibited significant antagonistic activity to several plant fungal pathogens, in particular to Magnaporthe oryzae, a fungal pathogen that causes highly destructive rice blast. Moreover, the authors identified several gene clusters through whole genome sequencing that encode for secondary metabolites. Further, they observed that strain GS-1 was able to produce the lipopeptide compounds, surfactin, fengycin, and plantazolicin, which showed inhibitory effects on M. oryzae. Underlying antagonistic mechanisms were derived from transcriptomics and metabolomics analysis.
The manuscript deals with the inhibition of fungus-based plant diseases that is a crucial agricultural problem through massive crop yield losses. Identification of novel fungizides are thus crucial to combat such diseases. Biocontrol using Bacillus species is attractive because they can produce spores, and a series of bioactive compounds to survive in adverse environmental conditions. Moreover, Bacillus species have many functions, which include promotion of plant growth and stress resistance. The present study combined different strategies to identify secondary metabolites of a newly isolated Bacillus strain by using cutting-edge omics-technologies. The authors were able to show by genomics, transcriptomics, metabolomics and functional screening that GS-1 inhibits fungal growth most likely through expressed secondary metabolite, which consequently reduce transcription of fungal metabolic genes. The manuscript is scientifically sound. In addition, the abstract appropriately summarizes the study. The introduction covers all necessary points to understand the following results and methods. Moreover, the complete manuscript successively describes literature information and further results, which makes the manuscript understandable throughout the text. The results section is clearly presented, and the results support the discussion/conclusion. Figures and tables nicely summarize and support the flow text. In total, I have only three minor comments (see below).
Minor comments:
Lines 106-109 (Figure 1): Species names have to be written in italics.
Line 151 (Figure 5): Please provide a higher-resolution figure and a more detailed legend.
Line 501: should be “centrifugation” instead of “centrifuged”
Author Response
Dear reviewer,
Thanks for your helpful comments. We have revised all the three comments in the manuscript. Please see the attachment.
Best.
Xianghui Zhang
